



# ENSO-Conditioned Weather Resampling Method for Seasonal Ensemble Streamflow Prediction

Joost V.L. Beckers[1], Albrecht H. Weerts[1,2], Erik Tijdeman[3] and Edwin Welles[4]

[1] Deltares, Delft, the Netherlands
[2] Department of Environmental Sciences, Wageningen University, the Netherlands
[3] Department of Hydrology, University of Freiburg, Freiburg, Germany
[4] Deltares USA Inc, Silver Spring, Maryland, USA

*Correspondence to*: Joost V.L. Beckers (Joost.Beckers@deltares.nl)

**Abstract.** Oceanic-atmospheric climate modes, such as El Niño Southern Oscillation (ENSO), are known to affect the local streamflow regime in many rivers around the world. A method is proposed to incorporate climate mode information into the Ensemble Streamflow Prediction (ESP) method for seasonal forecasting. The ESP is conditioned on an ENSO index in two steps. First, a number of original historical ESP traces are selected based on similarity between the index value in the historical year and the index value at the time of forecast. In the second step, additional ensemble traces are generated by a stochastic ENSO-conditioned weather resampler. These resampled traces compensate for the reduction of ensemble size in the first step and prevent degradation of forecast skill in sub-basins that are less affected by ENSO. The skill of the ENSO-conditioned ESP is evaluated over 50 years of seasonal reforecasts of streamflows in three sub-basins of the Columbia River basin in the Pacific Northwest. An improvement in forecast skill up to 10% is found.

## 1 Introduction

The Ensemble Streamflow Prediction (ESP) forecasting method is a common way to produce seasonal outlooks of river volumes. It is used by River Forecasting Centers of the National Weather Service (NWS-RFC) and other U.S. agencies (Druce, 2001; Pica, 1997; McEnery et al., 2005). The ESP uses historical time series of mean areal precipitation (MAP) and mean areal temperature (MAT) and considers these as representative of the local climate (Twedt et al., 1977; Day, 1985). The historical MAP and MAT series are used as meteorological forcing to a conceptual hydrologic model to generate an ensemble of streamflow forecasts. The number of ensemble traces is equal to the number of historical years because every trace corresponds to a particular historical year. The initial model state is the current state of the watershed of interest, which is obtained from an update run with data-assimilation of recent gauge data. Depending on the type of watershed and the time of year, the initial conditions can affect the streamflows for several months ahead (Wood and Lettenmaier, 2008; Li et al., 2009; Shukla and Lettenmaier, 2011; Yossef et al. 2013). This gives the ESP predictive ability over a climatological forecast, i.e. a distribution of historical streamflows (Franz et al., 2003).



Despite the great improvements in general circulation model (GCM)-based seasonal forecasting over the past decades (Leung et al., 1999; Wood et al., 2002; Clark and Hay, 2004; Wood et al., 2005; Wood and Lettenmaier, 2006; Yuan et al. 2015), the ESP method is still the current practice at most NWS-RFC. One of the reasons for this is that ESP uses the same type of meteorological input, i.e. historical MAP and MAT, as is typically used for calibration of the hydrologic models

(Pica, 1997). GCM input typically needs to be downscaled and bias-corrected before it can be applied to hydrological modeling at the sub-basin scale. A second reason is that the ESP allows for a sampling of non-meteorological variables, such as water demand, from the same historical years as the meteorological inputs. The fact that all variables are taken from the same historical year automatically preserves any cross-correlation between them, which is important for water resources planning.

In the original ESP, the historical MAP and MAT series represent the average climate, that is, every historical year is treated as an equally likely future scenario. In many regions, however, the local climate is known to be teleconnected to inter-annual to decadal fluctuations in oceanic-atmospheric circulation patterns, such as the El Niño-Southern Oscillation (ENSO) and Pacific Decadal Oscillation (PDO) (Ropelewski and Halpert, 1986, 1996; Kiladis and Diaz, 1989; Halpert and Ropelewski, 1992; Diaz and Markgraf, 2000; McCabe and Dettinger, 2002). These fluctuations, or climate modes, affect the streamflow

regime in U.S. rivers (Redmond and Koch, 1991; Kahya and Dracup, 1993; Dracup and Kahya, 1994; Piechota and Dracup, 1996; Piechota et al., 1997; Mantua et al., 1997; Beebee and Manga, 2004; Tootle et al., 2005; Tootle and Piechota, 2006; Lu et al., 2011; Gedalof et al., 2012).

The phase of most climate modes is quantified by climate indices that are evaluated and published monthly. Taking this information into account in streamflow forecasting could enhance its skill. Several methods have thus been developed to

incorporate climate index information into the ESP. They can be classified into pre- and post-processing schemes (Werner et al. 2004; an et al. 2010). In the pre-processing approach, the MAP and MAT ESP inputs are modified to match the predicted climate anomalies (Perica, 1998). Hay et al. (2009) applied a climate-mode-dependent adjustment of hydrologic model parameters. Another pre-processing alternative is to generate synthetic input time series by random resampling of monthly MAP and MAT from historical years that have similar climate index values (Werner et al., 2004).  Although some

improvement of forecast skill was reported, Werner et al. (2004) concluded that these pre-adjustment techniques are computationally cumbersome and less suited for operational usage than post-processing techniques. Kang et al. (2010) also found the post-processing schemes more effective than pre-processing schemes in a Korean case study.

In the post-processing approach, the ESP output, i.e. the ensemble of hydrographs, is transformed to incorporate climate mode information. One technique is to weigh the ensemble traces according to the similarity between climate indices in the

historical year and the year of forecast (Croley II, 1996, 2003; Stedinger and Kim, 2010; Madadgar et al., 2012; Najafi et al., 2012; Bradley et al., 2015). Instead of a weighting scheme, Hamlet and Lettenmaier (1999) used a selection of ESP traces according to a classification of historical years based on ENSO and PDO climate indices. Although their results showed an improved specificity of the ensemble forecast, the classification leads to a reduction of ensemble members, because the number of historical years in each class is obviously less than the original number of ensemble members. A reduction of





ensemble size generally leads to a degradation of the statistical properties of the ensemble forecast and to a reduction of forecast skill (Richardson, 2001; Ferro, 2007).

Although less obvious, this problem also arises in other ensemble post-processing schemes. The effective ensemble size is reduced by applying weights to ensemble members. To be effective, the information that is added to the ensemble by the

weighting should be in balance with the reduction of the forecast uncertainty (Weijs and van der Giesen, 2013). However, to obtain a coherent forecast for a large watershed, the forecasting must be done using a single set of weights for all sub-basins, although the influence of the climate modes may differ per sub-basin. A weighing scheme that produces good results for sub-basins that are influenced by a particular climate mode may not perform well for sub-basins that are less affected by this climate mode. The forecast skill for these latter sub-basins may be compromised by the weighting scheme. This problem has

been underexposed in previous studies. Najafi et al. (2012) mentioned the loss of forecast skill for smaller ensemble size and used a modified skill score to remove the effect (Weigel et al., 2007). This conceals the negative effect that a weighting scheme could have on quantile estimates for sub-basins that are less affected by climate modes.

In this study, an ESP conditioning method on climate mode information is described that produces a gain in forecast skill in sub-basins that are affected by climate modes, while avoiding a loss of skill in other sub-basins. The method is a

combination of pre- and post-processing. The post-processing involves a selection of traces from the original ESP. In a pre-processing, a number of new ensemble traces are generated by a monthly weather resampler. The newly generated traces augment the ensemble up to the original number of traces and all ensemble traces are weighted equally. This preserves the statistical properties of the ESP ensemble and avoids loss of forecast skill due to reduction of (effective) ensemble size.

The method is explained in detail in Sect. 2. The study region and the data used are described in Sect. 3. Sect. 4 includes the

results obtained applying the method to the study area and a forecast skill assessment relative to the standard ESP. Sect. 5 summarizes and concludes the paper.

## 2 Method

The proposed method consists of two parts: a *subsampler,* which selects ensemble members from the original ESP and a *resampler,* which generates additional ensemble members.

### 2.1 Subsampler procedure

The subsampler procedure is a k-nearest neighbor (k-NN) type scheme, similar to the schemes used by Werner et al (2004) and Najafi et al. (2012). The selection is based on similarity between the climate index value at the time of forecast and the value on the same day of a historical year. The selection can be based on a single climate index or on multiple indices. In the case of multiple indices the similarity criterion is the Euclidian distance in (multi-)index phase space. Weights can be applied

to each index-dimension to represent the relative importance of each index. The choice of indices and their optimal weights



will depend on the region of interest. A correlation analysis of climate index versus MAP/MAT is a straightforward way to find the strongest teleconnections.

The number of ESP traces to be selected by the subsampler needs to be optimized. By selecting fewer traces, the forecast becomes more specific, as only the historical years most similar to the present year are included in the forecast. However,

there is a trade-off between specificity and sampling error. With fewer years, the resolution of the ensemble decreases and the sampling error increases. This reduction of skill can be overcome by adding more ensemble members as is done in this study by using a resampler.

## 2.2 Resampler procedure

The resampler generates new ensemble members to augment the dismissed traces in the subsampler scheme. The new traces

are generated by a monthly weather resampler that is loosely based on a method developed by Brandsma and Buishand (1998). The resampler generates synthetic time series of precipitation and temperature by sampling from the historical record. Instead of using full historical years, as in the standard ESP, individual months from different historical years are sampled and assembled into new meteorological time series. The selection of historical months is conditioned on similarity between climate indices. A monthly resampling period is chosen to preserve the within-month temporal correlations and

because most climate indices are also defined on a monthly time scale. It is assumed that the resampled time series represent realistic and equally likely representations of future weather as the full historical years in the original ESP.

The resampling procedure is as follows.

1.    To initiate the sampling, the reference date is set to the time of forecast.

2.    A historical year is selected by probability sampling, where the probability of selecting year $y$ is a function of the

20        weighted Euclidian distance between the climate index values on the reference date $m_{i,r}$ and on the same day of a historical year $m_{i,y}$. A Gaussian-type distribution is adopted for this probability:

$$P_y = \frac{1}{N} \exp\left( -\sum_i w_i \left( m_{i,y} - m_{i,r} \right)^2 \right) \tag{1}$$

where $w_i$ is a factor that represents the importance of climate index $i$. $N$ is a normalization factor so the sum of all $P_y$ equals one.

3.    From the selected historical year $y$, a month of climate indices and MAP and MAT values is added to the newly generated time series.

4.    The new reference date is set to the selected date in the historical year plus one month and a new search (step 2) is started.

When going through the selection procedure, the same historical year can be selected several times in consecutive

resampling rounds. The year of the reference date even has the highest probability of being re-selected because it has the greatest similarity to the reference climate index. However, other historical years also have a non-zero probability of being





selected. Therefore, the resampled time series typically consist of resampled months from several historical years. The resampling procedure can be repeated with different random seeds to generate an ensemble of synthetic weather time series. The weights $w_i$ in Eq. (1) can have any positive value (also larger than 1). Their values determine not only the relative importance of the climate indices $i$ but also the stringency of the similarity criterion. The probability of selecting a historical

year with a similar climate index becomes larger for large $w_i$. This increases the persistence of the climate phase signal and its effect on the streamflow forecast. For small values of $w_i$, historical months that have quite different climate indices will be selected. Consequently, the climate phase signal is lost after a few resampling rounds.

A stringent similarity criterion will lead to the same historical years being selected every time. This will produce many similar or even identical traces that resemble full historical years. In order for the ensemble to accurately describe the

uncertainty distribution, more variation in the ensemble traces is needed, which is achieved by setting a less stringent similarity criterion. The choice for an appropriate similarity criterion is thus a trade-off between conservation of the climate phase signal and generating sufficient variation in the ensemble traces.

The weights $w_i$ for each index needs to be tuned to produce the required persistence of the climate signal and variation of ensemble traces at the relevant forecast lead times. Criteria that can be used for persistence are for example the difference

between climate indices in consecutive months and the autocorrelation function.  By adjusting $w_i$ and comparing the autocorrelation and month-to-month differences for the resampled time series, the optimal value is determined.

## 3 Study area and data

### 3.1 Study area

As a case study, the method was applied to seasonal streamflow forecasting at three projects (dams) on Columbia River

tributaries in the Pacific Northwest (PNW), listed in Table 1. The watersheds are located in the Cascade Range (see Fig. 1), where runoff is dominated by snowmelt. The typical annual pattern displays a build-up of snowpack in winter and snow melt and runoff in spring. Figure 2 shows the average and standard deviation of the monthly streamflows for the three projects. The flows are highest and have the most variation in the snow melt season (May-June).



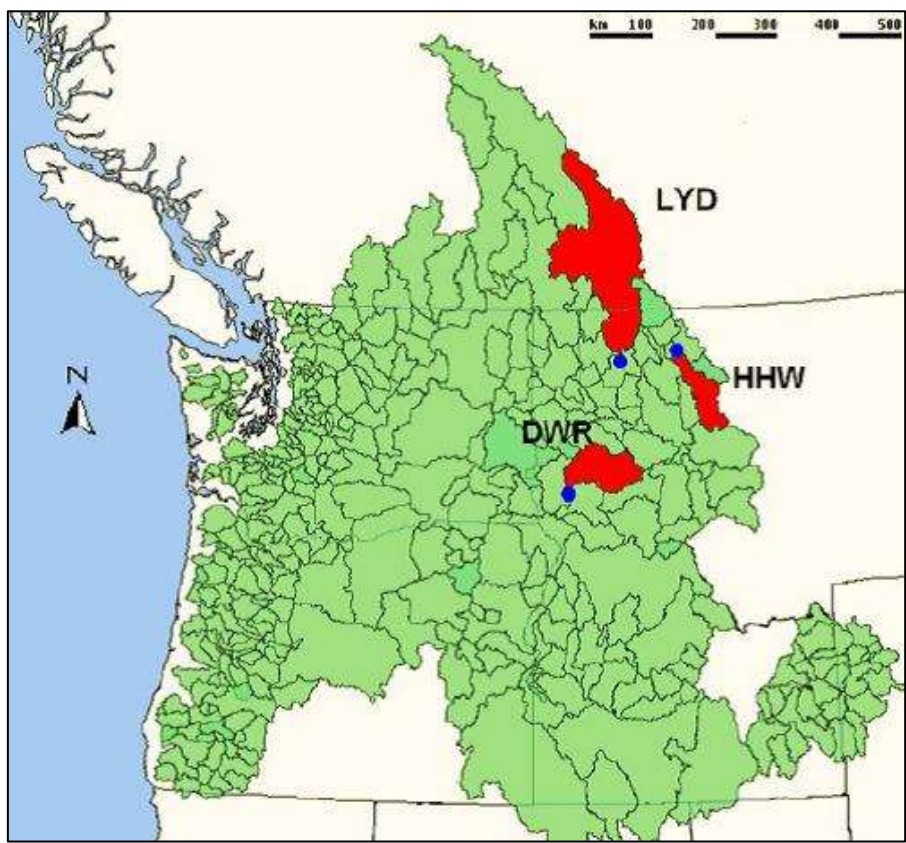

**Figure 1: Study area with the three test-sites and extent of sub-basins.**

One of the forecasting centres that use ESP for seasonal streamflow forecasting is Bonneville Power Administration (BPA). BPA is a self-financing federal agency based in Portland, Oregon that markets the hydroelectric power from 31 projects in

5   the Columbia River Basin (Bonneville Power Administration et al., 2001). The dams are operated following often competing needs and legal constraints, including hydropower production, supply of irrigation water, support of aquatic life and keeping the risk of undesirable peak flows and flooding at a minimum. Seasonal streamflow forecasting plays an important role in the dam operation planning and hydropower marketing. The high stakes on the energy market make even the smallest possible improvement in forecast skill worth pursuing.



**Table 1: Case study projects and properties.**

| Project | River | Drainage area (km$^2$) | Mean flow (m$^3$/s) | Powerhouse capacity (MW) |
|---|---|---|---|---|
| Libby Dam (LYD) | Kootenay | 23,270 | 310 | 600 |
| Hungry Horse (HHW) | Flathead | 4,145 | 100 | 428 |
| Dworshak (DWR) | Clearwater | 6,320 | 160 | 400 |

BPA uses an operational forecasting system called the Community Hydrologic Prediction System (CHPS) with ESP functionality for their seasonal streamflow outlooks (4 to 8 months lead time). The Sacramento Soil Moisture Accounting

model (SAC-SMA) (Burnash et al, 1973; Burnash, 1995) and SNOW-17 snow accumulation and ablation model (Anderson, 1976) are used for simulating and forecasting the hydrologic processes per sub-basin at a 6-hour time step, taking mean areal precipitation (MAP) and mean areal temperature (MAT) per sub-area as inputs. The conceptual sub-basin models were calibrated on 30 years of observational data. Initial (warm) states for the ESP forecasts are generated by running the models in operational mode, continuously blending in recent gauge data of e.g. snow pack and streamflow into the states.

The PNW climate is teleconnected with ENSO (Philander, 1990). The warm phase of ENSO (El Niño) is associated with warm and dry winters, whereas the cold phase (La Niña) has the opposite effect with colder and wetter than average winters (Ropelewski and Halpert, 1986; Redmond and Koch, 1991). Other climate phenomena have also been shown to influence the climate in the PNW (Lau and Sheu, 1988; Knight et al., 2006). The different climate modes may amplify or counteract each other, but each is considered to contain unique information that might have additional value for the streamflow predictions.

The influences of these climate phenomena make the PNW an interesting case study for the climate-conditioned ESP. Historical weather time series for the three sub-basins (6 hourly MAP and MAT) covering a period from 1949 to 2003 were provided by BPA. Historical values for a range of indices describing various climate modes were obtained from NOAA-CPC (http://www.cpc.ncep.noaa.gov/data/indices/).

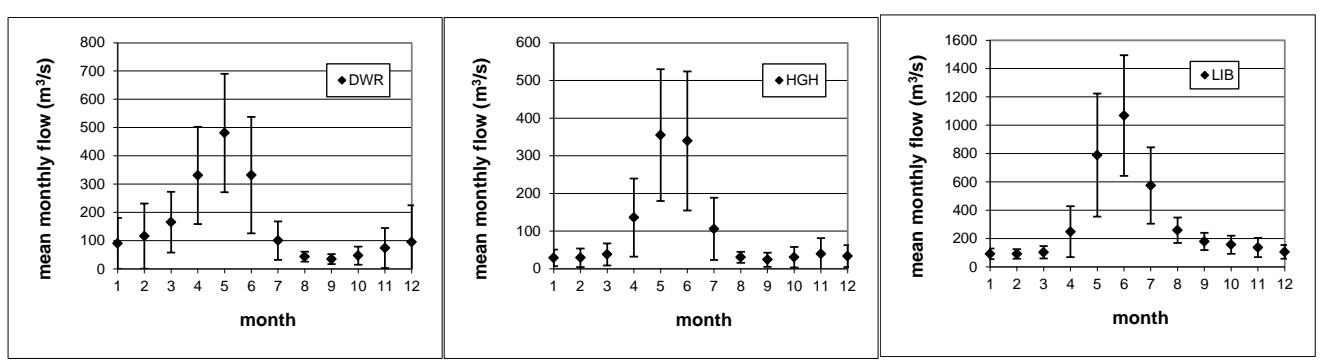

**Figure 2: Mean monthly streamflow and standard deviation for the test-basins Dworshak (DWR), Hungry Horse (HGH) and Libby Dam (LIB).**



## 3.2 Experimental Setup and Parameter Tuning

Several climate mode indices and combinations of indices for ensemble member selection and conditioning of the subsampler were evaluated, including the Pacific Decadal Oscillation (PDO), Multivariate ENSO Index (MEI), El Niño index NINO3.4 and Southern Oscillation Index (SOI). The MEI, as defined by Wolter (1998), showed the highest correlation with the historical streamflows in three sub-basins of interest and was therefore used for conditioning of the case study forecasts. The MEI combines several meteorological observables in a single metric and is issued monthly as a two-month value.

To tune the parameter $w$ for this case study, several values were evaluated. Figure 3 shows the distribution of differences between climate indices in consecutive months for the historical MEI series (1871-2013) and three resampled time series with $w$-values of 10, 25 and 100. From this figure, a value of $w$=100 seems optimal. However, the autocorrelation function (Fig. 4) shows that the $w$=100 series has a higher autocorrelation than the historical time series. This can be explained by the fact that the historical series has a 2-3 year quasi-biannual frequency (Barnett, 1991). The autocorrelation turns negative after 15 months lag time, indicating that a positive ENSO phase is most likely followed by a negative ENSO phase in the succeeding year and vice versa. This periodic behaviour cannot be reproduced by the basic lag-1 resampling method. The autocorrelation of the resampled time series simply decays to zero.

In order to approximate the persistence of the historical climate index series, a weight $w$ of 25 is chosen, which reproduces the autocorrelation of the historical MEI series at the relevant lead times for the seasonal forecasts, i.e. between 4 and 6 months.

The method was implemented as a module in Delft-FEWS, a hydrological forecasting and data management platform (Werner et al., 2013) upon which CHPS is built. The subsampler-resampler module was run from CHPS to generate meteorological forecasts with lead times up to 12 months for every month in the period 1949-2003. Next, ensemble streamflow reforecasts (forecasts in retrospect) were produced by running the hydrologic models, taking the subsampled and resampled MAP and MAT series as input. The year of reforecast was excluded from the subsampling and resampling schemes.



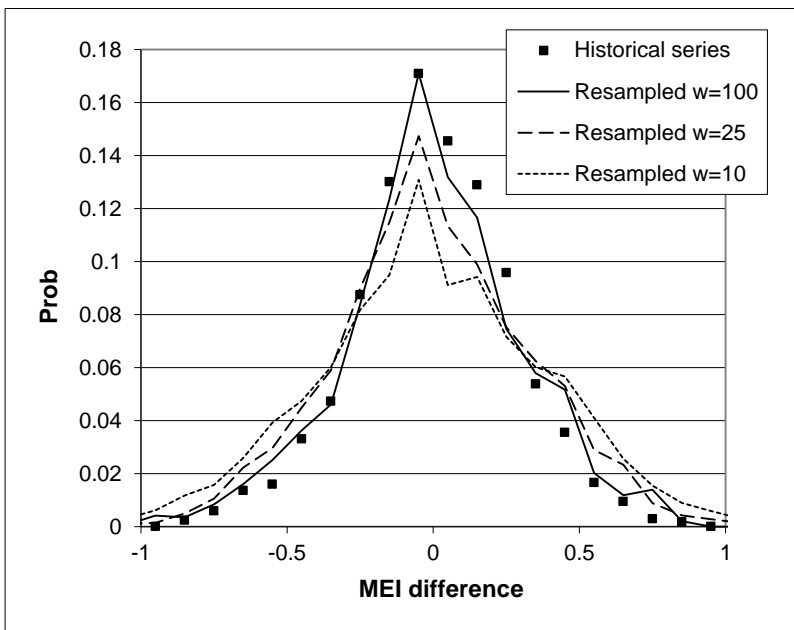

**Figure 3: Distribution of ENSO-MEI differences between consecutive months; historical series and three resampled time series with w-values of 10, 25 and 100.**

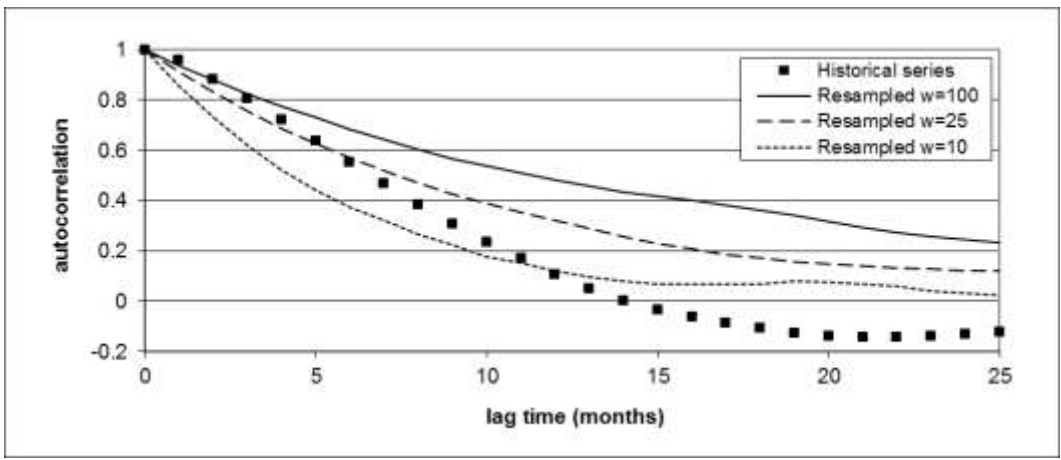

5  **Figure 4: Autocorrelation of ENSO-MEI signal for the historical and three resampled time series with w-values of 10, 25 and 100.**

Figure 5 shows example reforecasts of (a) climate index, (b) monthly mean precipitation, (c) monthly mean temperature and (d) monthly mean streamflow ensembles, starting from reference dates December 1$^{st}$ of 1973 (La Niña year), 1978 (neutral) and 1997 (El Niño year). The historical values are shown in red. Except for the shortest lead times in a few cases, the

10  historical traces fall within the range of the ensemble. The MEI-, precipitation- and temperature ensembles for the three starting dates differ due to the conditioning of the resampler. As a result, the streamflow ensembles have less spread than the original ESP and a better forecast skill, as will be shown in Sect. 4.




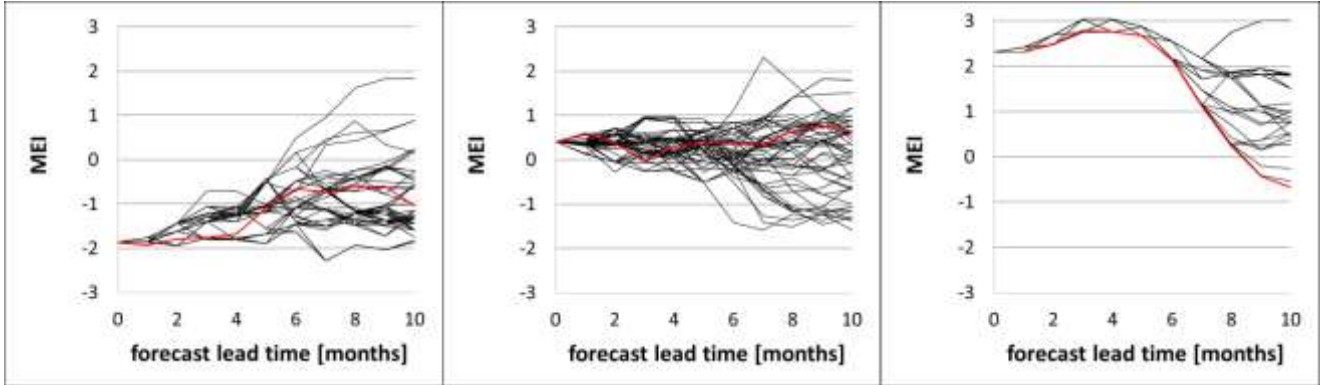

**Figure 5a: Resampled ensemble MEI forecasts. Forecast dates are December 1st 1973 (left), 1978 (middle) and 1997 (right). The historical time series are shown in red.**

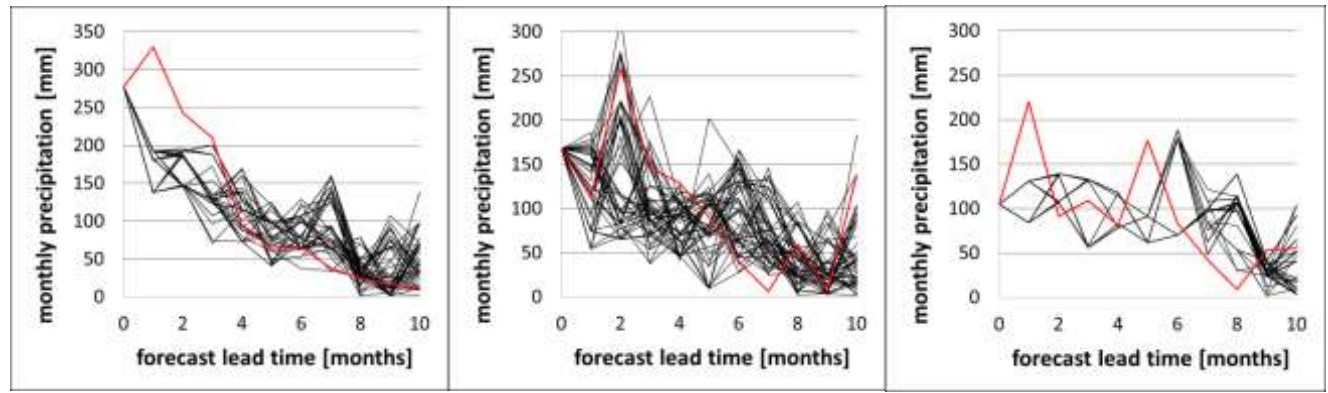

**Figure 5b: Resampled ensemble MAP forecasts for test-basin Dworshak. From left to right: forecasts from December 1st 1973 (left), 1978 (middle) and 1997 (right). The observed series are shown in red.**

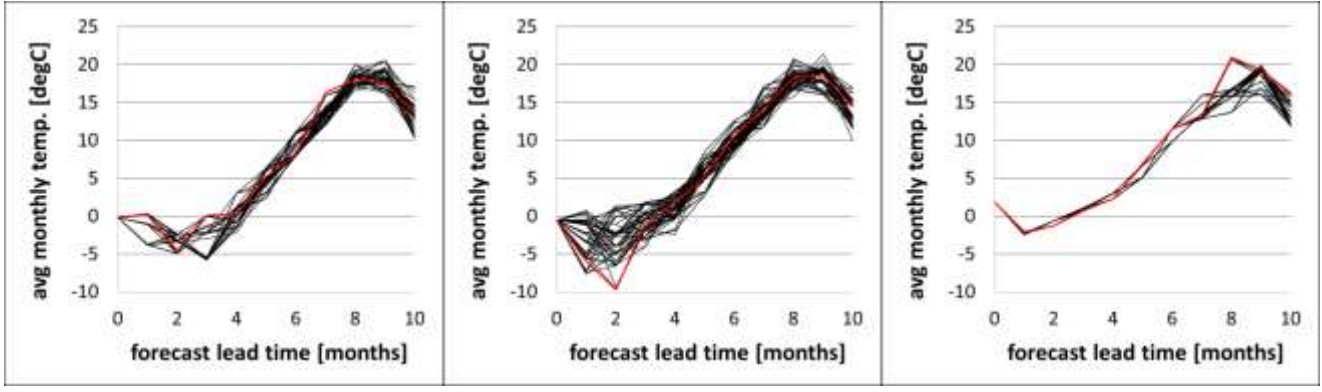

**Figure 5c: Resampled ensemble MAT forecasts for test-basin Dworshak. From left to right: forecasts from December 1st 1973**
10 **(left), 1978 (middle) and 1997 (right). The observed series are shown in red.**





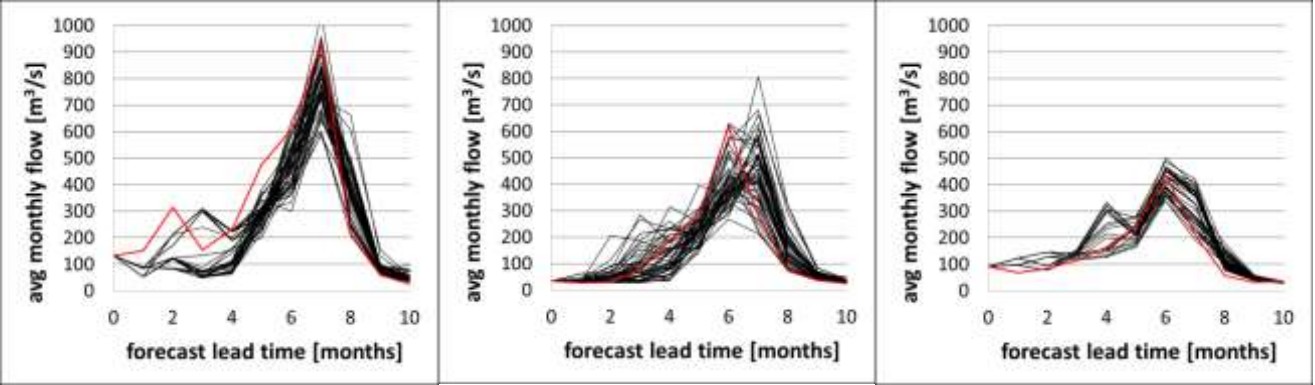

**Figure 5d: Resampled ensemble streamflow forecasts at test location Dworshak. Forecast dates are December 1st 1973 (left), 1978 (middle) and 1997 (right). The historical runs are shown in red.**

5   Figure 6 shows the number of unique ensemble members as a function of lead time. Different behaviour is found for the

three forecasts. The 1997 forecast starts off from a rather extreme positive MEI. The probability of resampling a different

historical year depends on the difference in MEI. Since the number of historical years that have such extreme MEI values is

limited, a small set of historical years gets re-sampled multiple times and the number of unique ensemble traces after 5

resampling rounds is only 17. In contrast, the 1978 forecast starts off from an average MEI value, with many historical years

10  with similar MEI values to resample from. As a result, each of the 50 ensemble traces is unique after 5 resampling rounds.

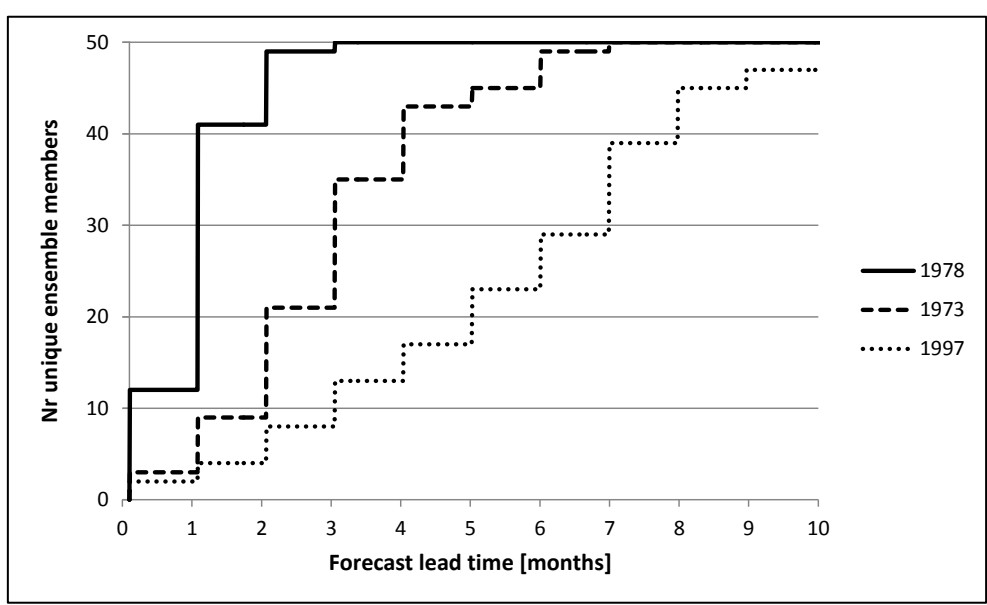

**Figure 6: The number of unique ensemble traces in 50-member ensembles of resampled time series (w=25), starting from December 1st, 1978, 1973 and 1997.**





### 3.3 Forecast evaluation

The skill of the forecasts was assessed in terms of Root Mean Square Error (RMSE) of the ensemble mean, Brier Score (BS) and Continuous Ranked Probability Score (CRPS). The RMSE is a direct measure of the accuracy of the mean forecast but it does not account for ensemble spread. The BS and CRPS are integral measures of ensemble forecast quality (Jolliffe and Stephenson, 2003; Wilks, 2006). The Brier score was computed for a threshold level at 80% exceedance probability of the monthly flow for each sub-basin.

The subsampler-resampler method was run in parallel to the original ESP method within CHPS, to enable a comparison. The skill metrics for the two methods were compared through relative skill scores, for example the Brier Skill Score (BSS):

$$\text{BSS} = 1 - \frac{\text{BS}_{model}}{\text{BS}_{reference}} \qquad (2)$$

Where the $BS_{reference}$ is the Brier Score of the standard ESP method. The skill metrics were calculated using the Ensemble Verification System (Brown et al., 2010). The next section focuses on forecast skill for streamflows in May and June. These

months have the most variation (see Fig. 2), which makes the effect of an improved forecast more pronounced.

### 4 Results

The performance of the subsampler selecting historical years from the original ESP based on climate mode similarity was first evaluated without the addition of resampled time series. Figure 7 shows the BSS as a function of number of ESP ensemble members. Skill scores reported here refer to June flow and are averaged over forecast lead times between 1 and 12

months. The 50-year ensemble is identical to the original ESP and has a BSS of 0. Upon reducing the ensemble size, the BSS increases for two of the three sub-basins (Dworshak and Hungry Horse) as a result of dismissing historical years with dissimilar MEI values. This indicates that the climate mode conditioning is shifting the ensemble forecast towards the most probable outcome.





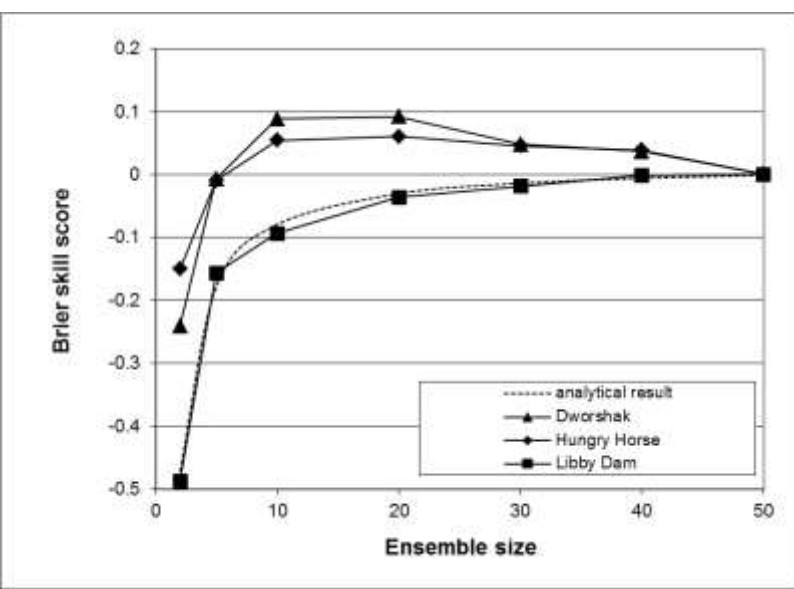

**Figure 7: Subsampler method: Brier Skill Score (80% threshold) of May-June streamflow forecasts as a function of number of historical ESP ensemble members.**

However, for one sub-basin (Libby Dam), the BSS decreases for smaller ensemble sizes. The reduction of the number of

ensemble members has a negative effect on its statistical properties. The sampling uncertainty increases, which counteracts the gain in forecast skill from the climate mode information. The dashed line represents the general behaviour of the BSS for a randomly reduced ensemble size, as described by Ferro (2007). Streamflows at Libby Dam have the weakest correlation with MEI. Apparently, the MEI information has little additional value for the Libby Dam streamflow forecasts and their skill follows this trend. For the other two basins the BSS also drops below zero for ensemble sizes less than 10.

Figure 8 shows the BSS for forecasts from the combined subsampler-resampler method, where the ensemble members that were dismissed in the subsampler are replaced by resampled traces. The ensemble size is thus 50 in all cases and the BSS is a function of the number of original ESP members (full historical years).  In contrast to Fig. 7, the BSS for all test basins are now positive over the full range. This demonstrates that the loss of skill from the reduction of ensemble size can be neutralized by additional ensemble traces from the resampler method. The improvement of skill in terms of RMSE and

CRPS was also investigated and found in agreement with the Brier skill score (results not shown). A mix of 10 historical years from the subsampler ESP and 40 additional resampled traces produces the best result for these sub-basins.





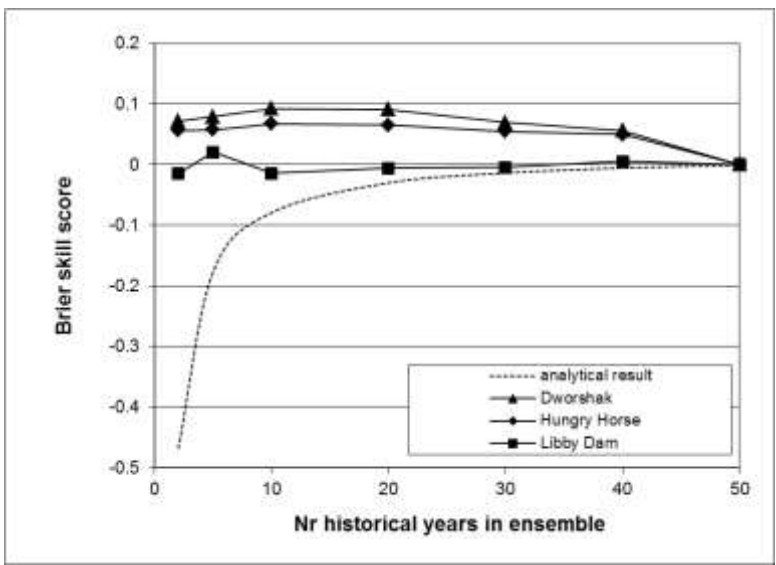

**Figure 8: Combined subsampler-resampler method (w=25): Brier Skill Score of May-June streamflow forecasts as a function of the number of historical ESP ensemble members. The subsampled historical ESP is augmented by resampled traces so the total ensemble size is always 50.**

5    Figure 9 shows the forecast skill as a function of forecast lead time. A combination of 10 historical and 40 resampled traces is used for all lead times. Three different skill metrics are shown for the June flow from the Dworshak sub-basin. The other two test basins show similar but less pronounced behaviour (results not shown). A positive skill is found up to 12 months of forecast lead time. This confirms the persistent nature of the ENSO climate mode. Because of this persistence, the conditioning of the subsampler and resampler on the climate phase at the time of forecast produces a positive skill over

10    several months up to a full year in the future.

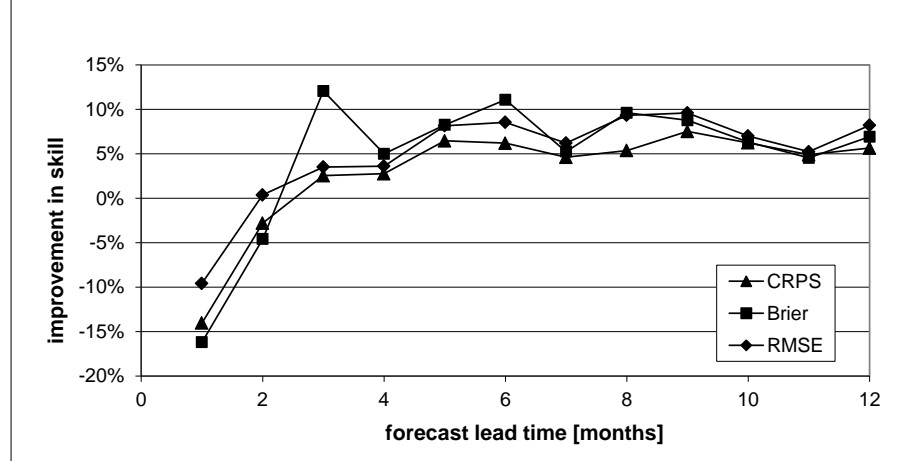

**Figure 9: Improvement in May-June streamflow forecast skill of the subsampler-resampler (w=25) method relative to the standard ESP as a function of lead time for DWR test site, three different skill metrics: CRPS, Brier Score and RMSE.**





Figure 9 shows that for lead times of 1 or 2 months, the skill is negative. This is due to a small effective ensemble size of the resampled traces for the shortest lead times, as discussed in Sect. 3.1. In order to maintain climate mode information on the seasonal time scale, the similarity criterion was set fairly stringent ($w$=25). This produces good results for the 4 to 6-month lead times, but it causes the same small set of historical years to be selected in the first resampling rounds every run.

Although the absolute number of ensemble members is 50, a small subset of historical years keep re-appearing in the resampled time series at the shortest lead times. This has a negative effect on the statistical properties of the ensemble and on the forecast skill. For longer lead times, this effect vanishes (see Fig. 6).

## 5 Discussion

The results in the previous section show that the subsampler-resampler method is able to improve the ESP forecast skill by 5

to 10% in two of the three test basins. This improvement seems modest compared to the 28% gain in forecast skill reported by Werner et al. (2004) and 27% by Bradley et al. (2015) who used similar post-processing methods. We note, however, that the performance may vary considerably per sub-basin. Werner et al. (2004) found a much smaller skill improvement of 4 and 6% for two other sub-basins, which is comparable to the results found in this study. Moreover, Werner et al. (2004) used a separate calibration of post-processing parameters per sub-basin. Many operational applications require equally weighted

ensembles for all sub-basins in the area of interest. This requirement does not allow for a per-sub-basin optimization.
For the third sub-basin in our case study, Libby Dam, no improvement of skill was found. The streamflows in this sub-basin have the lowest correlation with MEI and the local climate is least affected by ENSO. It was shown that dismissing ensemble members from the ESP leads to a reduction of forecast skill for this sub-basin because of the degradation of statistical properties of the ensemble. However, the additional traces from the resampler restore the forecast skill to that of the original

ESP. The adverse effect of the dismissal of ensemble traces by the subsampler is neutralized by the resampler traces. This is an important advantage of the subsampler-resampler method in operational settings, where avoiding loss of forecast skill anywhere is at least as important as improving the skill for a few sub-basins.
The subsampler-resampler method also has some practical advantages over alternative approaches. Firstly, the subsampler-resampler produces an equal-likelihoods streamflow ensemble, in contrast to the ensemble-weighting schemes. Also, the

total number of ensemble traces can be set equal to the original number of ESP members. This facilitates a comparison between the forecast skill of the conditioned ESP and that of the unconditioned ESP. Even more importantly, it facilitates the migration of an operational forecasting system from a standard ESP to a climate-mode conditioned ESP, since the downstream processes that use the streamflow ensemble as input need not be updated. Finally, the resampler method allows for a parallel sampling of non-meteorological variables from the historical record, with automatic preservation of cross-

correlations. This is an important advantage for agencies like BPA that use these variables (e.g. power demand) in their water resources planning tools.



There are several parameters in the subsampling-resampling method that must be reconsidered or recalibrated if the method is applied to other regions or lead times of interest. Firstly, the relevant climate modes should be identified for the region of interest. To simplify the test case in this study, we have used only a single climate index: MEI. Next, the number of original ESP traces to be selected in the subsampler should be set. The optimal number of traces was found to be 10 in the current

application, which is close to the values of 7 found by Werner et al. (2004), 12 by Najafi et al (2012) and 9 by Bradley et al. (2015). Apparently, a selection of 15% to 20% of original ESP traces gives the best performance for this type of ESP subsampling.

Another calibration parameter is the weight per climate index in the resampler procedure, which determines the persistence of the climate phase signal and the spread of the ensemble. It was found that a weight $w$=25 gave the best results for the 4- to

6-month lead times of interest in this case study, although it leads to an underdispersed ensemble for the shorter lead times. A less stringent similarity criterion, i.e. a smaller $w$, would improve the spread for short lead times. However, this would lead to a less persistent climate phase signal and loss of forecast skill for the longer lead times.

There are several opportunities for further improvement of the method. For the Columbia basin, a conditioning on other climate modes (e.g. PDO) could improve the forecast skill. This is being explored by BPA at the moment. The performance

at short lead times can possibly be improved by introducing a random time shift in the historical resampling scheme. This would introduce more variability in the resampled traces without compromising the persistence of the climate phase signal. Another possible improvement is to employ GCM-based climate mode forecasts instead of the lag-1 resampling procedure described in Sect. 2.2. This is left for future research.

## 5 Acknowledgements

This work was supported by Bonneville Power Administration. The authors wish to thank Ann McManamon from BPA for valuable comments and providing test data.

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
