# Peer review of "ENSO-Conditioned Weather Resampling Method for Seasonal Ensemble Streamflow Prediction"

_Hydrology and Earth System Sciences, 2016_

## Referee Comment (RC1) · Anonymous Referee #1 · 7 Mar 2016

Summary:

In this paper, the authors propose a technique that combines a post-processing step – i.e., sub-sampler of raw ensemble streamflow prediction (ESP) outputs based on climate index similarity – with a pre-processing step that generates synthetic precipitation and temperature time series via resampling, based on climate index similarity, to force hydrologic model simulations and re-populate the previously sub-sampled ensemble forecast. The method is applied in three catchments located in the Pacific Northwest, using the SAC-SMA and Snow-17 models, for seasonal (May-June) streamflow forecasting. The authors conclude that their framework is an improvement in skill (RMSE, Brier Score and Continuous Ranked Probability Score) over both standard ESP and

climate-based subsampling.

The paper is in general well written and well organized, the proposed technique is scientifically sound and the results are quite interesting. Further, the connection with the existing literature on this topic is nicely conducted. In my opinion, the manuscript has a lot of potential for publication in HESS, but the authors need to clarify some methodological choices, revise some statements, and include omitted results to show if the method is actually robust.

Major comments:

1. Why didn't the authors include the results for improvement in skill (as in Figure 9) for Libby and Hungry Horse? I think that showing the results at these locations is critical to demonstrate that the proposed technique is an advance over raw ESP and climate-based subsampling (see comment #14 for more details on this).

2. P4, L29: It is inferred from this paragraph that the reference date is set to the day when the forecast is initialized. Further, it is also mentioned that "the year of the reference date even has the highest probability of being re-selected". However, later in the paper the authors mention that "the year of reforecast was excluded from the subsampling and resampling schemes" (P8, L24). These statements are confusing, so the authors should clarify what was actually done. In my opinion, the year of the reference date (or initialization time) should NOT be included in the subsampling/resampling procedures, since that year is the one forcing the forecast.

3. P8, L2: The authors state that "several climate mode indices and combinations of indices for ensemble member selection and conditioning of the subsampler were evaluated". However, from the same paragraph it is implied that MEI was selected because it provided the highest correlation with historical streamflow. Did the authors actually test several combinations of climate indices? Moreover, it has been shown that PDO strongly affects interannual variability of runoff in this region (e.g., McCabe, G.J., Wolock 2014; Sagarika et al. 2015). Did the authors perform any experiments

including both MEI and PDO in the subsampling process? I think this manuscript would greatly benefit if - at least for the subsampler method - additional experiments showing the use of PDO were included. My guess is that the poor results obtained at Libby may be related to this issue.

Minor comments:

4. P1, L23: The authors should note that the hydrologic model does not necessarily have to be conceptual in ESP frameworks.

5. Throughout the manuscript: the authors refer to "reforecasts" or "forecasts in retrospect" when reporting results, but it might be better to use the word "hindcasting" (Beven and Young 2013).

6. P2, second paragraph: the text may be enriched by adding a few more references (Hamlet and Lettenmaier 1999; Tootle et al. 2007; Abudu et al. 2010; Sagarika et al. 2015).

7. P2, L18: Several studies recommend developing custom climate indices for the basin(s) of interest using reanalysis datasets (e.g., Grantz et al. 2005; Regonda et al. 2006; Block et al. 2009; Opitz-Stapleton et al. 2007; Bracken et al. 2010; Mendoza et al. 2014), instead of using standard climate indices for predicting seasonal runoff volumes. This point could be made in the introduction.

8. P2, L21: The reference is missing here.

9. P5, L17: A better title for section 3 would be "Example Application".

10. P7, Table 1: It would be more informative to add mean basin elevation (or elevation range), mean annual runoff and mean annual precipitation (mm/yr), and runoff ratio. I think that powerhouse capacity is not relevant here.

11. I strongly encourage the authors to improve the quality (resolution) of Figures 1, 4, 5, 7 and 8. This is critical to enhance the readability of the paper.

12. Figures 7 and 8: The authors could merge the results displayed here into a single figure, using different colors for different methods (for instance, red for subsampler, and black for combined subsampler-resampler), and keeping the title of x-axis label as "Number of historical years in ensemble". This would allow a direct comparison between the proposed method and the benchmark technique (i.e. only sub-sampling). I also think that the authors should add two additional panels (similar to the one described) with results of CRPSS – which is in my opinion a much more interesting score to assess the skill of ensemble systems – and RMSE. Further, it should be mentioned in the caption that results are averaged over lead times of 1-12 months.

13. Figures 7-9: The captions indicate that results are for May-June flows, but the text refer to June flows. What is actually being presented? If results are for May-June flows, are these aggregated (i.e. how many values are used for computing the scores, Nyears or 2 x Nyears)? Is the 80% flow computed from all monthly streamflow values, or only from May and June historical flows?

14. Figure 9: As pointed in comment #1, the authors are encouraged to add and discuss results for Libby and Hungry Horse in this figure. This could be done by or adding two panels (b and c, for instance), or extra lines with different colors for each basin. The improvement in skill could also be compared to that obtained from using only sub-sampling (the benchmark method) to understand the added value of re-populating the ensemble.

15. P13, L10-16: The authors might want to re-word or delete a couple of sentences. For instance, they point for Figure 8 that "in contrast to Fig. 7, the BSS for all test basins are now positive over the full range", which is NOT true for the Libby reservoir (there are still negative BSS values). Moreover, the authors mention that "a mix of 10 historical years from the subsampler ESP and 40 additional resampled traces produces the best result for these sub-basins", which is innacurate again when looking at Libby (higher BSS is obtained using five historical years).

[Figure]

Suggested minor edits:

16. P1 L23: "forcing" -> "forcings".

17. P2, L27: "case study" -> "case study basin".

18. P2, L26: "weigh" -> "weight".

19. P3, L19-21: "Sect." -> "Section".

20. P5, L13: "needs" -> "need".

21. P7, L9: "of e.g." -> "with"; "into the states" -> "into model states".

22. P8, L1: "parameter tuning" -> "parameter calibration".

23. P12, L18: "the most variation" -> "the largest variation".

References:

Abudu, S., J. P. King, and T. C. Pagano, 2010: Application of Partial Least-Squares Regression in Seasonal Streamflow Forecasting. J. Hydrol. Eng., 612–623.

Beven, K., and P. Young, 2013: A guide to good practice in modeling semantics for authors and referees. Water Resour. Res., 49, 5092–5098, doi:10.1002/wrcr.20393.

Block, P. J., F. A. Souza Filho, L. Sun, and H.-H. Kwon, 2009: A Streamflow Forecasting Framework using Multiple Climate and Hydrological Models. JAWRA J. Am. Water Resour. Assoc., 45, 828–843, doi:10.1111/j.1752-1688.2009.00327.x.

Bracken, C., B. Rajagopalan, and J. Prairie, 2010: A multisite seasonal ensemble streamflow forecasting technique. Water Resour. Res., 46, W03532, doi:10.1029/2009WR007965.

Grantz, K., B. Rajagopalan, M. Clark, and E. Zagona, 2005: A technique for incorporating large-scale climate information in basin-scale ensemble streamflow forecasts. Water Resour. Res., 41, W10410, doi:10.1029/2004WR003467.

Hamlet, A. F., and D. L. Lettenmaier, 1999: Columbia River Streamflow Forecasting Based on ENSO and PDO Climate Signals. J. Water Resour. Plan. Manag., 125, 333–341.

McCabe, G.J., Wolock, D. M., 2014: Spatial and temporal patterns in conterminous United States streamflow characteristics. Geophys. Res. Lett., 41, 6889–6897, doi:10.1002/2014GL061980.Received.

Mendoza, P. A., B. Rajagopalan, M. P. Clark, G. Cortés, and J. McPhee, 2014: A robust multimodel framework for ensemble seasonal hydroclimatic forecasts. Water Resour. Res., 50, 6030–6052, doi:10.1002/2014WR015426.

Opitz-Stapleton, S., S. Gangopadhyay, and B. Rajagopalan, 2007: Generating streamflow forecasts for the Yakima River Basin using large-scale climate predictors. J. Hydrol., 341, 131–143, doi:10.1016/j.jhydrol.2007.03.024.

Regonda, S. K., B. Rajagopalan, M. Clark, and E. Zagona, 2006: A multimodel ensemble forecast framework: Application to spring seasonal flows in the Gunnison River Basin. Water Resour. Res., 42, W09404, doi:10.1029/2005WR004653.

Sagarika, S., A. Kalra, and S. Ahmad, 2015: Interconnections between oceanic–atmospheric indices and variability in the U.S. streamflow. J. Hydrol., 525, 724–736, doi:10.1016/j.jhydrol.2015.04.020. http://www.sciencedirect.com/science/article/pii/S0022169415002693.

Tootle, G. A., A. K. Singh, T. C. Piechota, and I. Farnham, 2007: Long Lead-Time Forecasting of U.S. Streamflow Using Partial Least Squares Regression. J. Hydrol. Eng., 442–451.
* * *

---

## Referee Comment (RC2) · Anonymous Referee #2 · 14 Mar 2016

This paper presents a two-pronged approach for conditioning ESP forecasts on ENSO conditions. In the first step, a sub-sample of ESP forecasts are selected from an ensemble (e.g. of size 50) by conditioning on a climate index. This reduces the number of ensemble members. In the second step, the ensemble is augmented to the original size by sampling precipitation and temperature from the historical record, conditioned on the climate index, and thereafter producing additional ESP forecasts.

I think the paper presents a pragmatic approach to incorporating climate information into ESP forecasts and for enlarging the ensemble size. These types of technique are of wide interest in the hydrologic ensemble forecasting community.

The writing is generally of publication quality but several figures need improvement.

[Figure]

I have some issues with the clarity, execution and explanation of the science. If the authors can thoroughly address the issues, some of which are not simple, my opinion is the paper should eventually be published in HESS.

General comments 1) A number of parameters are tuned on the basis of subjective analysis for the whole period of interest. Because this is a forecasting paper, the parameter values ought to be determined from an objective analysis that can then be cross-validated using a leave-out scheme. If the results are not cross-validated then the results are potentially inconclusive. Given that the results are marginal, and perform best for the period tuned to (4–6 months lead time), I suggest this is quite important. Ideally, the following elements would be cross-validated: a. The climate index selection b. The number of optimal ESP sub-samples selected c. The "weight", w. If cross-validation isn't used, justification is required.

2) The results use the Brier score (for 80% exceedance probability forecasts) and CRPS as probabilistic measures. I think the paper would be much stronger if accuracy skill and reliability results were separated. Whether skill is attributable to accuracy or reliability or both may vary significantly with lead time. Also, it is stated repeatedly throughout the paper that a small effective number of ensemble members is associated with "degradation of the statistical properties" of the ensemble forecast. What exactly does this mean? I suggest be specific and explain exactly which properties are affected and how they are affected. This is particularly important in the results (P15 L6) and discussion (P15 L18–19).

3) The resampling approach performs poorly for short lead times. Particularly, as shown by Figure 9, the forecasts at short lead times are up to 16% worse. The resampler produces much too narrow forecasts for the first couple of months. This is a problem with the ad-hoc nature of the approach, the spread in the ensembles at any given lead time could be either too narrow or too wide or somewhere in between. What happens if the resampling begins several months prior to the forecast date (i.e. lag 2 or lag 3 MEI)? It's a hard sell to say that forecasts get worse as lead time shortens. At
what point should the forecasts be ignored? I encourage a resolution.

Specific comments 4) Abstract, last sentence: This needs to explicitly say when and where improvements of up to 10% are found and probably should also say that the results for short lead times are worsened.

5) P4 L1 suggests selecting climate indices based on correlations with MAT/MAP. But P8 L4–5 reports MEI was selected on the basis of correlation with streamflow. Please make more consistent.

6) MEI is a two-month index. Were two-month values of the other indices considered?

7) Equation (1): The summation appears to be the squared Euclidean distance (no square root). Also, how are indices in different units handled (is it implicitly through scaling/weighting)?

8) Figure 2: It might be better to show percentile intervals rather than statistics based on normal distributions (unless of course the data is very normal).

9) P13 L10–13: The BSS is marginally negative for some cases for Libby Dam, so the statement saying BSS is positive for all cases needs correcting. Also, re the comment about Figure 8, the text says the BSS is a function of "number of the original ESP members", but I think it means the number of sub-sampled years (hence less than 50 on the x-axis is Figure 8).

10) The introduction states that section 5 summarises and concludes the paper, but section 5 is headed "Discussion". Suggest renaming.

11) P15 L10 should say in two of the test basins *at lead times greater than X*

12) P15 L13–15: Operational applications should be flexible enough to adapt to different methods if there's a proven benefit. So this argument doesn't carry a lot of weight.

13) P16 L13–14: I'm confused by this. PDO was apparently investigated already in this study and disregarded.

Technical corrections (typing errors, etc.) 14) Figure 2 and Table 1. Abbreviations do not match for Hungry Horse and Libby Dam.

15) P12 L19 and elsewhere: Text refers to June flow instead of May–June flow.

16) There are some instances of weigh and weighing instead of weight and weighting. Will be easy to find and correct.

17) Improve the figure quality. Many are blurry.

18) Is Figure 5 one figure or four? There are four captions.

―――――――――――――――――

---

## Editor Comment (EC1) · M.-H. Ramos (Editor) · 18 Mar 2016

Dear authors, your manuscript has received two referee comments, which are crucial, in my opinion, to improve the quality of your paper. I encourage you to submit your responses as soon as possible to enhance the discussions and allow the evaluation of the manuscript for potential revisions and eventually its final publication in HESS. More specifically, I have the following additional comments: 1) consider submitting the figures with an improved quality as supplement material together with your replies to the referee comments. This is important to allow us to correctly evaluate their value with respect to the scientific results of the paper; 2) clarify the role of the parameter w. In this regard, section 3.2 deserves to be improved with further details on the procedure adopted; 3) please, also check the requirements for additional results on the studied catchments (those not shown in the current version of the paper) and other months/seasons. This could bring robustness to your conclusions. We are looking forward to your reply and contribution to the open discussion. Best regards, Maria-Helena Ramos.

---

## Author Response (AR1)

**ENSO-Conditioned Weather Resampling Method for Seasonal Ensemble Streamflow Prediction**

Joost V.L. Beckers, Albrecht H. Weerts, Erik Tijdeman and Edwin Welles

5  This memo includes a point-by-point response to the comments by the Editor and two Anonymous Referees. For each point, we indicate the changes that were made to the manuscript. At the end of this document is a marked-up manuscript version showing all changes.

**RESPONSE TO EDITOR'S COMMENTS**

10  Editor Decision: Reconsider after major revisions (02 May 2016) by Maria-Helena Ramos

Comments to the Author:

The authors are acknowledged for the answers to the two referees and are encouraged to submit a revised version of their manuscript. Please, specifically, consider the major remarks of the referees, namely:

- include omitted results to show if the method is actually robust;
15  This has been done, see Figure 7 and 8.

- provide more evidence that "several climate mode indices and combinations of indices for ensemble member selection and conditioning of the subsampler were evaluated" as mention in Section 3.2 by the authors;
The aim of this paper is to explain the method and demonstrate its use in a simple test case: a limited number of
20  test basins and a single climate index (MEI). It is shown how the method performs for two test basins where the streamflow is correlated with MEI and for a test basin that is not strongly affected by the climate signal, in this case Libby. An optimization of the method for a larger number of test basins using other climate signals (including PDO) will be done by BPA. This is mentioned in the discussion. Results of that optimization study may be published separately at a later time.

- clarify how parameters were tuned or determined, indicating when cross validation was performed or not (and explaining the reasons behind the choices made);
The parameter setting is explained in Sect. 3.2. We add the following additional information:
- The climate index MEI was chosen from several candidates (SOI, ENSO3.4, PDO) based on a
30     correlation analysis. The correlation analysis was done between the index values in December and the annual flow volume in the next year for the period 1948-2008. The MEI has the highest correlation (around 0.5 for DWR and HHW, 0.35 for LYD) with the historical streamflows for the three sub-basins and was therefore selected for this case study. PDO has the second highest correlation (around 0.35 for all three sub-basins). The correlation analyses were repeated for different months (lead times) and for
35     different historical periods and the results showed a consistent picture that MEI is the strongest teleconnection at these lead times and PDO is the second strongest.
- It is also explained in section 3.2 how the value of the weight w was determined: based on the autocorrelation of the MEI signal (Figures 3 and 4). It is explained why a value of 25 was found appropriate for forecasting at the seasonal time scale.
40  - The choice for the number of ESP sub-samples is based on heuristic techniques. Several values were tested (see Figure 7). From this Figure, a combination of 10 subsampled members gives the best skill, but larger values also produce a positive skill. In Sect. 5, the optimal value of 10 sub-samples is

compared to values found in comparable methods from literature: "The optimal number of traces was found to be 10 in the current application, which is close to the values of 7 found by Werner et al. (2004), 12 by Najafi et al (2012) 5 and 9 by Bradley et al. (2015). Apparently, a selection of 15% to 20% of original ESP traces gives the best performance for this type of ESP subsampling."

5   A cross-validation using split datasets was not done because the resampling method relies on a large historical dataset. Splitting the dataset into a calibration and validation set would increase the uncertainty and reduce the skill. The results from a split dataset would therefore not be comparable to the results for the full dataset. Also note that none of the studies mentioned above (Werner et al., 2004; Najafi et al, 2012; Bradley et al., 2015) includes a cross validation of parameter settings.

 - provide details on the methodology, as stressed by Referee 2;
More details were given where requested by Referee 2.

 - provide enhanced figures for the results, such as these can be better evaluated.
15   This was done.

**RESPONSE TO COMMENTS FROM REFEREE #1**

**Anonymous Referee #1**

5    In this paper, the authors propose a technique that combines a post-processing step – i.e., sub-sampler of raw ensemble streamflow prediction (ESP) outputs based on climate index similarity – with a pre-processing step that generates synthetic precipitation and temperature time series via resampling, based on climate index similarity, to force hydrologic model simulations and re-populate the previously sub-sampled ensemble forecast. The method is applied in three catchments located in the Pacific Northwest, using the SAC-SMA and Snow-17 models, for
10   seasonal (May-June) streamflow forecasting. The authors conclude that their framework is an improvement in skill (RMSE, Brier Score and Continuous Ranked Probability Score) over both standard ESP and climate-based subsampling.

     The paper is in general well written and well organized, the proposed technique is scientifically sound and the
15   results are quite interesting. Further, the connection with the existing literature on this topic is nicely conducted. In my opinion, the manuscript has a lot of potential for publication in HESS, but the authors need to clarify some methodological choices, revise some statements, and include omitted results to show if the method is actually robust.

20   Major comments:
     1. Why didn't the authors include the results for improvement in skill (as in Figure 9) for Libby and Hungry Horse? I think that showing the results at these locations is critical to demonstrate that the proposed technique is an advance over raw ESP and climate-based subsampling (see comment #14 for more details on this).
     Figures for Hungry Horse and Libby have been added as additional frames in Figure 8 (formerly Figure 9). For
25   Hungry Horse the improvement in skill is smaller than for Dworshak. For Libby, there is no gain or loss in forecast skill.

     2. P4, L29: It is inferred from this paragraph that the reference date is set to the day when the forecast is initialized. Further, it is also mentioned that "the year of the reference date even has the highest probability of
30   being re-selected". However, later in the paper the authors mention that "the year of reforecast was excluded from the subsampling and resampling schemes" (P8, L24). These statements are confusing, so the authors should clarify what was actually done. In my opinion, the year of the reference date (or initialization time) should NOT be included in the subsampling/resampling procedures, since that year is the one forcing the forecast.
     The year of hindcast is excluded from the resampling to be able to assess the forecasting skill. At the start of the
35   resampling procedure, the reference date is equal to the forecast date. That year is excluded from the resampling, so a different historical year must be selected in the first resampling round. In the next resampling round, however, the reference date is set to a date in the historical year that was selected in the first round. That year is not excluded from the resampling, so it can be selected again.

40   3. P8, L2: The authors state that "several climate mode indices and combinations of indices for ensemble member selection and conditioning of the subsampler were evaluated". However, from the same paragraph it is implied that MEI was selected because it provided the highest correlation with historical streamflow. Did the authors actually test several combinations of climate indices? Moreover, it has been shown that PDO strongly affects interannual variability of runoff in this region (e.g., McCabe, G.J., Wolock 2014; Sagarika et al. 2015). Did
45   the authors perform any experiments including both MEI and PDO in the subsampling process? I think this manuscript would greatly benefit if - at least for the subsampler method - additional experiments showing the use of PDO were included. My guess is that the poor results obtained at Libby may be related to this issue.
     The aim of this paper is to explain the proposed method and demonstrate its use in a simple test case: a limited number of test basins and a single climate index. This proof of concept includes demonstrating how the method

performs for a test basin that is not strongly affected by the climate signal, in this case Libby. An optimization of the method for other locations in the Columbia River basin and using other climate signals (including PDO) will be done by BPA. This is mentioned in the discussion. Results of that optimization study may be published separately at a later time.

Minor comments:

4. P1, L23: The authors should note that the hydrologic model does not necessarily have to be conceptual in ESP frameworks.

Agreed. We removed the word 'conceptual'.

5. Throughout the manuscript: the authors refer to "reforecasts" or "forecasts in retrospect" when reporting results, but it might be better to use the word "hindcasting" (Beven and Young 2013).

We changed 'reforecasts' into 'hindcasts' (4 instances). The first time that the term 'hindcast' is mentioned, we add '(reforecasts)' in brackets for clarity. The term 'reforecasts' is also used in literature, e.g. by Werner (2004) and Wood (2002).

6. P2, second paragraph: the text may be enriched by adding a few more references (Hamlet and Lettenmaier 1999; Tootle et al. 2007; Abudu et al. 2010; Sagarika et al. 2015).

Thanks for this suggestion. We added these references.

7. P2, L18: Several studies recommend developing custom climate indices for the basin(s) of interest using reanalysis datasets (e.g., Grantz et al. 2005; Regonda et al. 2006; Block et al. 2009; Opitz-Stapleton et al. 2007; Bracken et al. 2010; Mendoza et al. 2014), instead of using standard climate indices for predicting seasonal runoff volumes. This point could be made in the introduction.

We feel that these custom climate indices should be part of the optimization of the method for a specific area and lead time of interest. Our paper focuses on explaining the basic method and demonstrating its use in a simple test case of three locations and a single climate index. Optimization of the method for a larger study area using multiple indices and/or custom climate indices would be a separate study. BPA is currently carrying out the optimization and results of that may be published at a later time (see also our answer to point 3).

8. P2, L21: The reference is missing here.

This was corrected.

9. P5, L17: A better title for section 3 would be "Example Application".

Agreed, we changed the title.

10. P7, Table 1: It would be more informative to add mean basin elevation (or elevation range), mean annual runoff and mean annual precipitation (mm/yr), and runoff ratio. I think that powerhouse capacity is not relevant here.

Agreed. We added average elevation, mean runoff, precipitation and runoff ratio and removed powerhouse capacity.

11. I strongly encourage the authors to improve the quality (resolution) of Figures 1, 4, 5, 7 and 8. This is critical to enhance the readability of the paper.

The figures were improved.

12. Figures 7 and 8: The authors could merge the results displayed here into a single figure, using different colors for different methods (for instance, red for subsampler, and black for combined subsampler-resampler), and keeping the title of x-axis label as "Number of historical years in ensemble". This would allow a direct comparison

between the proposed method and the benchmark technique (i.e. only sub-sampling). I also think that the authors should add two additional panels (similar to the one described) with results of CRPSS – which is in my opinion a much more interesting score to assess the skill of ensemble systems – and RMSE. Further, it should be mentioned in the caption that results are averaged over lead times of 1-12 months.

Figures 7 and 8 were combined and two additional panels with CRPS and RMSE results were added.

Results are averaged over lead times 3 to 12 months, because the skill for 1 and 2 months is poor. The fact that the skill scores are averaged is mentioned in the caption.

13. Figures 7-9: The captions indicate that results are for May-June flows, but the text refer to June flows. What is actually being presented? If results are for May-June flows, are these aggregated (i.e. how many values are used for computing the scores, Nyears or 2 x Nyears)? Is the 80% flow computed from all monthly streamflow values, or only from May and June historical flows?
What is shown are the verification scores for forecasts of monthly streamflows for May and June. This was clarified in the text.

14. Figure 9: As pointed in comment #1, the authors are encouraged to add and discuss results for Libby and Hungry Horse in this figure. This could be done by or adding two panels (b and c, for instance), or extra lines with different colors for each basin. The improvement in skill could also be compared to that obtained from using only subsampling (the benchmark method) to understand the added value of re-populating the ensemble.
Additional panels were added to Figure 8 (formerly Figure 9) for Libby and Hungry Horse and results are discussed in the text. The gain in forecast skill for these subbasins is less than for Dworshak. For Libby there is no gain in forecast skill.

15. P13, L10-16: The authors might want to re-word or delete a couple of sentences. For instance, they point for Figure 8 that "in contrast to Fig. 7, the BSS for all test basins are now positive over the full range", which is NOT true for the Libby reservoir (there are still negative BSS values). Moreover, the authors mention that "a mix of 10 historical years from the subsampler ESP and 40 additional resampled traces produces the best result for these sub-basins", which is inaccurate again when looking at Libby (higher BSS is obtained using five historical years).
The small negative score for Libby and the positive skill for five historical years are attributed to uncertainty/noise in the calculation, i.e. statistical uncertainty related to the limited number of hindcasts. We rephrased these sentences to:
"in contrast to the skill of the subsampler forecasts, the subsampler-resampler produces in general a positive skill over the full range. The marginal loss of skill for Libby is attributed to statistical uncertainty of the skill score calculation."
"a mix of 10 historical years from the subsampler ESP and 40 additional resampled traces produces in general the best result for these sub-basins"

Suggested minor edits:
16. P1 L23: "forcing" -> "forcings".        Agreed, changed.
17. P2, L27: "case study" -> "case study basin".  Agreed, changed.
18. P2, L26: "weigh" -> "weight".        Agreed, changed.
19. P3, L19-21: "Sect." -> "Section".        No change made, we think this is HESS-style
20. P5, L13: "needs" -> "need".        Agreed, changed.
21. P7, L9: "of e.g." -> "with"; "into the states" -> "into model states". Rephrased to:
'… blending in recent snow pack and streamflow gauge data into model states'
22. P8, L1: "parameter tuning" -> "parameter calibration".  Agreed, changed.
23. P12, L18: "the most variation" -> "the largest variation". Agreed, changed.

**RESPONSE TO COMMENTS FROM REFEREE #2**

**Anonymous Referee #2**

This paper presents a two-pronged approach for conditioning ESP forecasts on ENSO conditions. In the first step, a sub-sample of ESP forecasts are selected from an ensemble (e.g. of size 50) by conditioning on a climate index. This reduces the number of ensemble members. In the second step, the ensemble is augmented to the original size by sampling precipitation and temperature from the historical record, conditioned on the climate index, and thereafter producing additional ESP forecasts. I think the paper presents a pragmatic approach to incorporating climate information into ESP forecasts and for enlarging the ensemble size. These types of technique are of wide interest in the hydrologic ensemble forecasting community. The writing is generally of publication quality but several figures need improvement.

I have some issues with the clarity, execution and explanation of the science. If the authors can thoroughly address the issues, some of which are not simple, my opinion is the paper should eventually be published in HESS. General comments:

1) A number of parameters are tuned on the basis of subjective analysis for the whole period of interest. Because this is a forecasting paper, the parameter values ought to be determined from an objective analysis that can then be cross-validated using a leave-out scheme. If the results are not cross-validated then the results are potentially inconclusive. Given that the results are marginal, and perform best for the period tuned to (4–6 months lead time), I suggest this is quite important.

Ideally, the following elements would be cross-validated:
a. The climate index selection
b. The number of optimal ESP sub-samples selected
c. The "weight", w. If cross-validation isn't used, justification is required.

We believe that the parameter setting is not subjective. It is explained in the manuscript how the climate index selection was done and how the weight $w$ was determined on statistical analysis of the climate signal before the actual hindcasts were made, i.e. without using hindcast information:
    a.  The climate index MEI was chosen from several candidates (SOI, ENSO3.4, PDO) based on a correlation analysis with historical streamflow data. We added a sentence "A correlation analysis was done between the index values in December and the annual flow volume in the next year." to clarify how this was done. The MEI has the highest correlation (around 0.5 for DWR and HHW, 0.35 for LYD) with the historical streamflows for the three sub-basins and was therefore selected for this case study. PDO has the second highest correlation (around 0.35 for all three sub-basins). These correlation analyses were repeated for different months (lead times) and for different historical periods and the results showed a consistent picture that MEI is the strongest teleconnection at these lead times and PDO is the second strongest.
    b.  See below.
    c.  It is explained in section 2.3, page 8 how the value of the weight $w$ was determined, based on the autocorrelation of the MEI signal (Figures 3 and 4). It is explained why a value of 25 was found suitable for forecasting at the seasonal time scale.

For the third parameter: the number of ESP sub-samples selected, several values were tested and results presented in Figure 7. A consistent positive forecast skill is found for two sub-basins, except for less than 10 sub-samples. The reason for the poor scores for small numbers of sub-samples is explained in the text (see also

response to remark nr 2). The absence of a gain in forecast skill for the third sub-basin and the loss of forecast skill for short lead times are also explained.

Based on Figure 7, a combination of 10 subsampled members and 40 resampled members is chosen as optimal in this case, but larger values also produce a positive skill. In Sect. 5, the optimal value of 10 sub-samples is compared to values found in comparable methods from literature: "The optimal number of traces was found to be 10 in the current application, which is close to the values of 7 found by Werner et al. (2004), 12 by Najafi et al (2012) 5 and 9 by Bradley et al. (2015). Apparently, a selection of 15% to 20% of original ESP traces gives the best performance for this type of ESP subsampling." Note that none of these studies included a cross validation of parameter settings.

A cross-validation on split datasets indeed could provide insight into the uncertainty of the results. However, the uncertainty of the calculated verification scores would increase for a smaller dataset, so we are not sure if this analysis would be conclusive. In general, we feel that we have shown that the results are rational and robust to the choice of parameter settings.

2) The results use the Brier score (for 80% exceedance probability forecasts) and CRPS as probabilistic measures. I think the paper would be much stronger if accuracy skill and reliability results were separated. Whether skill is attributable to accuracy or reliability or both may vary significantly with lead time. Also, it is stated repeatedly throughout the paper that a small effective number of ensemble members is associated with "degradation of the statistical properties" of the ensemble forecast. What exactly does this mean? I suggest be specific and explain exactly which properties are affected and how they are affected. This is particularly important in the results (P15 L6) and discussion (P15 L18–19).
In answer to the first remark, we use three different skill metrics that are quite common in forecasting. They are related to typical usage of a probabilistic forecast, namely the best estimate or mean forecast (the accuracy of which is measured by RMSE), the probability of exceeding a critical threshold (measured by Brier score) and the overall reliability of the forecast probabilities (measured by CRPS). Many other skill scores and measures of forecast quality are possible but we feel that these three cover the most important aspects of a probabilistic forecast.

In answer to the second remark, the effect of a reduction of ensemble size on verification scores is well-known. The effect of ensemble size on Brier score has been analysed extensively by Richardson (2001) and Ferro (2007). An extension to CRPS was done by Ferro et al (2008). The RMSE of the ensemble mean also increases with decreasing ensemble size (see e.g. Ho et al., 2013, Eqn (1) or Weigel 2007 for weighted ensembles). An ensemble of fewer members has a less accurate ensemble mean and is less well capable of accurately describing a probability distribution. In the manuscript, we explain this effect qualitatively and refer to existing literature where appropriate:

- Page 3: "A reduction of ensemble size generally leads to a degradation of the statistical properties of the ensemble forecast and to a reduction of forecast skill (Richardson, 2001; Ferro, 2007; Ferro et al, 2008)."
- Page 4: "…there is a trade-off between specificity and sampling error. With fewer years (ensemble members), the resolution of the ensemble decreases and the sampling error increases."
- Page 12, Line 20 and further: "The reduction of the number of ensemble members has an adverse effect on its statistical properties. The sampling uncertainty increases, which counteracts the gain in forecast skill from the climate mode information. The dashed lines represent the general behaviour of the forecast skill for a randomly reduced ensemble size, as described by Ferro (2007) for BSS. The analytical results for CRPSS and RMSE were derived from Ferro et al (2008), Eqn. 22 and Ho et al. (2013), Eqn. 1 respectively."
- Page 15 (discussion): "It was shown that dismissing ensemble members from the ESP leads to a reduction of forecast skill for this sub-basin that is similar to the expected reduction for a randomly reduced ensemble, as described by Richardson (2001), Ferro (2007) and Ferro et al. (2008). "

More references (Ferro et al. 2008; Ho et al. 2013) were added for effects on CPRS and RMSE as these scores were added to Figure 7 following a suggestion from reviewer nr 1. We believe that the general description of the effect and references to literature are adequate for this manuscript.

3) The resampling approach performs poorly for short lead times. Particularly, as shown by Figure 9, the forecasts at short lead times are up to 16% worse. The resampler produces much too narrow forecasts for the first couple of months. This is a problem with the ad-hoc nature of the approach, the spread in the ensembles at any given lead time could be either too narrow or too wide or somewhere in between. What happens if the resampling begins several months prior to the forecast date (i.e. lag 2 or lag 3 MEI)? It's a hard sell to say that forecasts get worse as lead time shortens. At what point should the forecasts be ignored? I encourage a resolution.

To make full use of the information of the current climate signal we do not recommend starting the resampling several months prior to the forecast date. Instead, we describe a way to improve the performance at shorter lead times in Sect. 5: "The performance at short lead times can possibly be improved by introducing a random time shift in the historical resampling scheme. This would introduce more variability in the resampled traces without compromising the persistence of the climate phase signal. "

The poor performance at short lead times is not necessarily problematic if the ESP is used only for forecasting at longer lead times (4 months or longer) and other techniques (e.g. NWP weather input) are used for forecasting at short lead times.

Specific comments

4) Abstract, last sentence: This needs to explicitly say when and where improvements of up to 10% are found and probably should also say that the results for short lead times are worsened.

The forecast skill improvement of 5 to 10% for two sub-basins is mentioned as well as the lack of improvement for the third sub-basin. We choose not to mention the poor performance for short lead times in the abstract because the method is meant for seasonal forecasting at longer lead times, as the title says. The poor performance at 1 and 2 month lead time is discussed extensively in the results and discussion sections. A possible solution is described on Page 16, Line 15-16.

5) P4 L1 suggests selecting climate indices based on correlations with MAT/MAP. But P8 4–5 reports MEI was selected on the basis of correlation with streamflow. Please make more consistent.

We changed page 4 line 1 to "historical streamflows". MAP/MAP would also be possible but that is not what was done here.

6) MEI is a two-month index. Were two-month values of the other indices considered?

Indeed two- and more-month averaged values of other indices were considered in the correlation analysis, but the results were no better than for the original indices.

7) Equation (1): The summation appears to be the squared Euclidean distance (no square root). Also, how are indices in different units handled (is it implicitly through scaling/weighting)?

The indices can be normalised or the weights $w$ could carry units. In principle, the weights can have any positive value (as mentioned on Page 5, Line 3).

8) Figure 2: It might be better to show percentile intervals rather than statistics based on normal distributions (unless of course the data is very normal).

Agreed, the figure was changed to show median and 10% and 90% error bars.

9) P13 L10–13: The BSS is marginally negative for some cases for Libby Dam, so the statement saying BSS is positive for all cases needs correcting. Also, re the comment about Figure 8, the text says the BSS is a function of "number of the original ESP members", but I think it means the number of sub-sampled years (hence less than 50 on the x-axis is Figure 8).

The number of the original ESP members is equal to the number of sub-sampled years

10) The introduction states that section 5 summarises and concludes the paper, but section 5 is headed "Discussion". Suggest renaming.
We changed the outline in the introduction to: "Sect. 5 discusses the results."

11) P15 L10 should say in two of the test basins *at lead times greater than X*
We changed this sentence to: "… by 5 to 10% in two of the test basins for lead times greater than 2 months."

12) P15 L13–15: Operational applications should be flexible enough to adapt to different methods if there's a proven benefit. So this argument doesn't carry a lot of weight.
Operational applications typically require a coherent seasonal forecast over the entire basin. A separate calibration per sub-basin may affect the spatial correlations between the sub-basins.

13) P16 L13–14: I'm confused by this. PDO was apparently investigated already in this study and disregarded.
PDO was considered, but MEI was found to have a better overall correlation with the streamflows in the three sub-basins (see Sect. 3.2). Therefore, MEI was used in the single index example application, but PDO would be the first candidate in an extension to multivariate conditioning.

Technical corrections (typing errors, etc.):
14) Figure 2 and Table 1. Abbreviations do not match for Hungry Horse and Libby Dam.
Agreed, changed accordingly.

15) P12 L19 and elsewhere: Text refers to June flow instead of May–June flow.
Agreed, changed accordingly.

16) There are some instances of weigh and weighing instead of weight and weighting. Will be easy to find and correct.
Two instances found and changed accordingly.

17) Improve the figure quality. Many are blurry.
Agreed, changed accordingly.

18) Is Figure 5 one figure or four? There are four captions.

Changed Figure 5 to one figure and one caption.

**References**

[revised manuscript text omitted]

---

## Author Response (AR2)

Editor's remarks:

- Page 2, line32-33: please, consider checking the expression "an improved specificity of the ensemble forecast". It is not fully clear to me what you mean by that.

'Specificity' is the term that is used by Hamlet and Lettenmaier (1999) for 'the relative size of the area between the upper and lower bound of the ensemble'. We prefer to use the same terminology when referring to their results.

- Page 4, line 10: "loosely based on…" Can you be more precise about the differences between the original proposal and your adaptation of the method referenced? As it is, the sentence is very vague and not much informative for the reader.

We added: "… loosely based on a method for daily rainfall resampling developed by Brandsma and Buishand (1998)". This should make clear that our resampling technique is similar to the method referenced, but the sampling interval (daily vs monthly) and the variable (rainfall vs climate index) are different.

- Page 4, lines 15-16: please, check the use of the English language in this part of the sentence: "… represent realistic and equally likely representations of…"

Changed to: "It is assumed that the resampled time series are realistic representations of future weather patterns and that they are equally likely to occur as the full historical years in the original ESP."

- Page 4, line 30-31: If I understand well, the year of reference (which is the year associated with the time of forecast) is used in the re-sample. This would however not be feasible operationally (as one would not have the data available for the time steps after the time of forecast). I am however not sure I understood it correctly because on page 8, line 24, you mention that "The year of hindcast was excluded…". Could you please check these two sentences and make then clearer for the reader?

The reference date is not the time of forecast/hindcast. The reference year changes with every resampling round, except when the same year is resampled twice. We rephrased the description of step 4 in the stepwise description of the algorithm and added an example:

1. To initiate the sampling, the reference date is set to the time of forecast.
   …
4. A new reference date is set by advancing one month and replacing the year by the selected historical year. For example, if the first reference date was January 1st 2016 and the selected historical year is 1997, the new reference date will be February 1st 1997. Subsequently, we proceed with the next resampling round and search for a historical year that is similar to the new reference date (step 2).

- Page 7, line 3: when you mention CHPS, are you making reference to the CHPS used by NOAA too? http://www.nws.noaa.gov/ohd/hrl/chps/ The way it is mentioned in the paper, it makes the reader think that CHPS is a specific system developed at BPA. Maybe if you could add a reference here, the ambiguity would disappear.

The NWS-CHPS and BPA-CHPS use the same application software (CHPS) but they are not the same system. The implementation of CHPS at BPA is specific to BPA. We rephrased the sentence and added a reference to a paper that describes the CHPS application. Hope this makes it more clear.

- Page 7, line 8: can you explicitly indicate what the period (years) is used for calibration?

Done

- Page 8, line 5: consider changing "in the next year" to "in the following year"

Done

- Page 8, line 6: consider changing "in three sub-basins of interest" to "in the three sub-basins of this study"

Done

- Page 8, line 10: considering the period 1871-2013, is it a stationary period? Aren't there trends observed during this period? How does non-stationarity impact your study and method? I think that maybe a sentence or two about the dependence (or not) on the hypothesis of stationarity of historic time series used in the methodology should be added to the paper (for instance, in the "Discussion" section).

Stationarity is assumed in standard ESP, which may indeed be an issue for some applications. However, it is not the aim of our method to address or resolve this. Our method aims to improve the ESP forecast skill by taking into account climate mode information, not climate change.

Furthermore, the period 1871-2013 is only used to calculate the graphs in Figs 3 and 4. The ESP and resampling/subsampling method use data from 1949-2003. Non-stationarity should be less of a problem for this shorter period.

- Figure 3 and Figure 4: I guess these results pertain to a specific sub-basin of the study area. Is that right? In that case, could you indicate that in the caption? Also what is the difference between "ENSO-MEI difference", "ENSO-MEI signal" and "MEI difference"? You use these in the figures. If they refer to the same thing, maybe it would be better to choose one unique term to refer to the same thing. Please check terminology (see also remark below)

MEI is a global indicator. It is not bound to a specific region. We changed "ENSO-MEI" into "MEI".

- Captions of Figure 1, Figure 2, and Figure 5: you use three different terminologies for the same object: "test-sites", "test-basins" and "test location". Could you please check throughout the paper and make sure that you use the same terminology for the same objects that you make reference to? This makes it much easier for a reader to follow the reasoning and the results.

We replaced 'test-sites', 'test/sub-basins' and 'test locations' by 'forecasting stations' throughout the text.

- Page 9, line 7 and Page 11, line 1: please, consider mentioning also in the text the sub-basin to which Figure 5 and Figure 6 refer to. This information should also be added in the caption of Figure 6.

Added 'at forecasting station Dworshak' to the text. Figure 6 is not specific for any forecasting station. The subsampling/resampling is done for all stations collectively.

- Figure 6: consider changing the caption to "Number of…". Also, please, check for the use of a unique term here too: "ensemble members" or "ensemble traces". It is better to choose one and stick to it all over the text.

Changed the caption to "Number of…".

Changed 'member' to 'trace' throughout the text.

- Page 11, line 19: check the use of punctuation (comma) here.

Removed the comma.

- Page 11-12: explain what you call "relative reduction in RMSE" (introduced on Page 12, line 7) already in the sub-section 3.3.

Added a sentence to section 3.3: "Likewise, the Continuous Ranked Probability Skill Score (CRPSS) and the relative improvement in RMSE are evaluated."

- Page 12, lines 6-8: check the use of the English language. I think some words (ex., "the") are missing here.

Added " … as a function of the number of ESP ensemble members."

- Page 13, line 9: are you sure it is "… attributed to statistical uncertainty of the skill score calculation"? How did you compute that? Otherwise, please moderate it with "probably" or "might be due to".

Changed to "…probably due to …" and added "(which could be verified by bootstrapping, but this is left for future studies)."

- Page 15, line 2: I think you mean "Sect. 3.2" instead of "Sec. 3.1". Please, check.

Indeed. Corrected

- Page 15, line 10: consider changing to "… in two of the three sub-basins …"

Done

- Page 15, line 14: "… used a separate calibration of post-processing parameters per sub-basin". I do not understand this sentence. Please check to make it clearer.

Changed to: "Werner et al. (2004) used a separate calibration of post-processing parameters to arrive at a different set of weights for each test station."

- Page 15, line 21: consider changing to "…for very small sub-samples."

Done

- Page 15, line 32: consider changing to "…as input do not need to be updated. Finally,…"

Done

- Page 16, lines 2-3: Is "water demand" stationary over historic periods? Can we use "historic data" to infer water demand of today? I think you could discuss some limitations also of using historic data for today's inferences. Water demand may have changed with societal changes (for instance, with growing concerns towards water economy). Again, I think that "stationarity" issues and implications to the methods presented would deserve a few lines in the discussion section.

I can imagine that BPA makes a correction of historical water demand to today's circumstances, but that is beyond the scope of our study. We simply note that parallel sampling of secondary variables is common practice in operational settings and that our method accommodates for this, in contrast to some other methods.

 - Page 15-16: Discussion section: I miss a discussion on the issues of cross-validation, i.e., taking some years out from calibration and then using them for validation. Could you add a sentence or two in the discussion section about it? In your opinion, how it may play a role and be considered in further studies?

There is hardly any calibration needed in our method. The only parameter that is calibrated on hindcast results is the number of ESP sub-samples. Figure 7 shows that a positive forecast skill is found for a range of choices for this parameter (for Dworshak and Hungry Horse), except for less than 10 sub-samples. Based on Figure 7, a value of 10 subsampled members is chosen for subsequent experiments, but larger values also produce a positive skill.

A bootstrapping procedure could provide insight into the uncertainty of the verification scores. We added a suggestion to apply a bootstrapping to assess the uncertainty in a future study on Page 14, lines 11-12.

- Finally, I think it is strange to have a paper without a "Conclusion" section. Could you add a section with the main conclusions of the study for the sake of completeness of the paper and clarity of the message and highlights of your results? Maybe it could be derived from a re-organization of the "Discussion" section.

We renamed the last section to "Discussion and conclusions".

 **Reviewer#2 remarks:**

1. P2, L2-5: The papers by Tootle, Abudu and Sagarika should not be cited here, since they do not relate to GCM-based seasonal forecasting. Instead, these should be included in L15.

Done

 2. Section 2.2: The authors may agree or not on this, but after reading this section several times, I still find difficult to understand how the resampler technique works. In my opinion, a comprehensive diagram would help a lot to clarify the method, and therefore help other readers to reproduce it.

We tried to clarify this by including an example in step 4: historical year selection and setting of a new reference date. Hope this helps.

3. Figure 1: I encourage the authors to add latitude, longitude and a scale bar.

Done

4. Table 1: Can the authors please add in the caption the period used to compute mean flow, precipitation and runoff ratio?

Done

5. P13, L8-9: The authors state that "the marginal loss of skill for Libby is attributed to statistical uncertainty of the skill score calculation". Can the authors clarify what does this mean? So far, no uncertainty (using bootstrapping, for instance) has been considered in the calculation of any score, so this statement seems misleading. Perhaps delete?

Changed to "…probably due to …" and added "(which could be verified by bootstrapping, but this is left for future studies)."

6. P15, L2: "as discussed in Sect. 3.1". Should it be Sect. 3.2?

Indeed, corrected.

7. P16, L18: Can the authors please clarify what they mean with "introducing a random time shift"?

Added an example for clarification: 'For example, instead of sampling a historical period April 1 – April 30, we shift 5 days back and sample March 27 – April 25. '

Suggested minor edits

8. P1 L12: "a number of… are" -> "a number of… is" (noun is singular, not plural).

Not adopted. From the dictionary: "Although the expression 'a number' is strictly singular, the phrase 'a number of' is used with plural nouns (as what grammarians call a determiner (or determiner)). The verb should therefore be plural."

9. P1 L17: "in the Pacific Northwest" -> "in the U.S. Pacific Northwest" (same in P5, L20).

Done

10. P3 L9: "latter" -> "later".

Not adopted. From the dictionary:
"Latter: relating to the second of two groups or things mentioned. "
"Later:  at some time subsequent to a given time."

11. P3 L16: "In a pre-processing" -> "In a pre-processing step".

Done

12. P6 L4: Add a comma after "Oregon".

Done

13. P15 L24: "resampler" -> "resampled".

Done